# Comparison of the triglyceride-glucose index and triglyceride-glucose-body mass index for predicting non-alcoholic fatty liver disease in elderly diabetic patients

Gaohui Zhu[1], Yihui Qu[2], Kanan Chen[2], Dingfa He[3], Jing Wang[2], Ziwei Tang[1], Xinyi Wang[1], Minqiao Zhang[4], Peilan Jiang[5], Ruijie Zhang[6], Kedan Cai📵[2]*

1 Department of Endocrinology, Ningbo Zhenhai Hospital of Traditional Chinese Medicine, Ningbo, Zhejiang, China, 2 Department of Nephrology, Ningbo No.2 Hospital, Ningbo, Zhejiang, China, 3 Zhaobaoshan Residential District Community Health Service Center, Ningbo, Zhejiang, China, 4 Department of Nephrology, The First People's Hospital of Xiangshan, Ningbo, Zhejiang, China, 5 Department of Endocrinology, The First People's Hospital of Xiangshan, Ningbo, Zhejiang, China, 6 Guoke Ningbo Life Science and Health Industry Research Institute, Ningbo, Zhejiang, China

* caikedan@ucas.edu.cn

## Abstract

### Background

Non-alcoholic fatty liver disease (NAFLD) is posing a challenge to global health systems. Developing an effective, simple, noninvasive method to identify NAFLD in elderly diabetic patients, track disease progression, and monitor treatment effects is importance. This study aims to assess the value of triglyceride-glucose index (TyG) and triglyceride-glucose-body mass index (TyG-BMI) in detecting NAFLD elderly patients with diabetes mellitus (DM).

### Methods

This study enrolled 6,882 individuals aged 60 years or older with DM who underwent liver ultrasonography in this cross-sectional study at Zhaobaoshan Residential District Community Health Service Center, from Jan. 1, 2015, to Oct. 19, 2023. And data was accessed for research purposes after Oct. 1,2024. Participants were randomly divided into a training group and a validation group in a 7:3 ratio. The diagnostic values of TyG and TyG-BMI were assessed using the area under the receiver-operating characteristic curve (AUROC) and Decision Curve Analysis (DCA). Two cut-off points were selected to rule out or rule in NAFLD, and we explored their specificity, sensitivity, negative predictive value, and positive predictive value.

### Results

There were 2,210 and 927 participants with NAFLD in the training and validation groups. In a fully adjusted model, TyG and TyG-BMI were correlated with an

**Data availability statement:** All relevant data are within the manuscript and its Supporting Information files.

**Funding:** This study was supported by Zhejiang Provincial Natural Science Foundation of China (LY20H05005), Medical Scientific Research Foundation of Zhejiang Province, China (2023KY283, 2024XY039), and Xiangshan County Science and Technology Plan Project - Research Fund Project(2023C6014). The funders had no role in study design, data collection and analysis, decision to publish, or preparation of the manuscript.

**Competing interests:** The authors have declared that no competing interests exist.

increased risk of NAFLD in the training group (TyG: OR=3.920, P<0.001; TyG-BMI: OR=1.032, P<0.001). These results were consistent in the validation group. The AUCs of TyG and TyG-BMI indicated that both had predictive value for NAFLD, with TyG-BMI showing the higher predictive accuracy. DCA suggested that TyG-BMI is preferable in clinical settings for both groups. In the training group, with a TyG-BMI cut-off of 212.886, the sensitivity was 80.6%, specificity 57.5%. With a cut-off of 251.741, the sensitivity was 32.6%, specificity 90.7%. Thus, a TyG-BMI < 212.886 could rule out NAFLD (SE = 80.6%, NPV = 77.8%), while a TyG-BMI ≥ 251.741 could rule in NAFLD (SP = 90.7%, PPV = 74.8%). These findings were similar in the validation group, with a TyG-BMI < 212.886 ruling out NAFLD (SE = 80.0%, NPV = 77.3%) and a TyG-BMI ≥ 251.741 ruling in NAFLD (SP = 91.5%, PPV = 76.4%).

## Conclusions

In conclusion, TyG-BMI is more accurate than TyG in predicting NAFLD in elderly participants with diabetes. This simple, non-invasive, and cost-effective tool effectively classifies elderly diabetic patients with and without NAFLD.

## 1. Introduction

Non-alcoholic fatty liver disease (NAFLD) is characterized by the excessive accumulation of triglycerides within hepatocytes and represents a growing burden on global healthcare systems. It affects approximately 25% of the adult population worldwide, with reported prevalence rates ranging from 6% to 35% across different studies [1–3]. Recent research has revealed notable age-related patterns in NAFLD prevalence, demonstrating a stepwise increase among males younger than 50 years and females younger than 60 years [4]. In contrast, the prevalence tends to decline in older populations [4], suggesting that the disease may exhibit distinct clinical features or risk profiles in elderly individuals compared to younger adults.

NAFLD encompasses a broad spectrum of liver conditions, ranging from simple steatosis to non-alcoholic steatohepatitis (NASH), which may further progress to cirrhosis and hepatocellular carcinoma [5,6]. Although liver biopsy remains the gold standard for diagnosing NAFLD [7], its invasive nature, associated risks, and high costs limit its routine use, particularly among older adults. Liver ultrasonography serves as a commonly employed non-invasive alternative; however, its diagnostic accuracy is highly dependent on the operator's skill and the quality of the imaging equipment. Moreover, ultrasound may fail to detect hepatic steatosis when fat infiltration is below 20% or in individuals with morbid obesity [8]. Therefore, there is an urgent need for a reliable, simple, and non-invasive approach to identify NAFLD, monitor disease progression, and evaluate therapeutic responses.

NAFLD is strongly associated with insulin resistance (IR), a key pathogenic factor in its development [9]. IR results in impaired glucose uptake by insulin-sensitive tissues, leading to elevated fasting blood glucose (FBG) levels and increased secretion of very low-density lipoprotein (VLDL). Notably, NAFLD and type 2 diabetes mellitus

(T2DM) frequently coexist. Approximately 25% of patients with NAFLD are also diagnosed with T2DM, while NAFLD is present in nearly 75% of individuals with T2DM [10]. The bidirectional relationship between these conditions exacerbates the risk of metabolic complications and advanced liver disease. Furthermore, the presence of age-related comorbidities, particularly those involving lipid metabolism and fibrosis, may further aggravate NAFLD in elderly patients. Accordingly, early identification of elderly individuals with T2DM who are at high risk for NAFLD is of critical importance.

In recent years, several surrogate indices have been developed to estimate IR. Among them, the triglyceride-glucose (TyG) index, calculated from fasting triglycerides (TG) and FBG, has gained widespread clinical adoption due to its ease of use and practicality [11,12]. Studies suggest that the TyG index may outperform the traditional homeostasis model assessment of insulin resistance (HOMA-IR) in certain settings [13]. A modified version, the TyG-body mass index (TyG-BMI), incorporates additional anthropometric and metabolic parameters, offering improved reflection of IR, particularly in non-diabetic individuals [14]. TyG-BMI has also demonstrated superior diagnostic performance for NAFLD in women and in younger and middle-aged populations, as evidenced by higher area under the curve (AUC) values [4]. However, whether the TyG index or TyG-BMI is more effective for identifying NAFLD in elderly patients with T2DM remains unclear. This study, therefore, aims to determine which of these two indices provides better predictive value for detecting NAFLD in elderly patients with diabetes.

## 2. Methods

### 2.1. Study subjects

The research protocols adhered to the principles of the Declaration of Helsinki and were approved by the Ethics Committee of Ningbo Zhenhai Hospital of Traditional Chinese Medicine (Ethics Approval Number: p-2024–003). This study utilized retrospective, de-identified data from elderly patients with diabetes, and all datasets used for analysis had been stripped of personally identifiable information, making it impossible to trace the data back to individual participants. The analysis posed minimal risk to subjects. In accordance with relevant national ethics guidelines, including Article 5(1), Section 5, Chapter 2 of the 2023 Guidelines for the Establishment of Ethics Committees for Clinical Research Involving Humans, the Ethics Committee granted a waiver of informed consent. Furthermore, the original data collection process included general consent clauses permitting the use of anonymized clinical data for future scientific research.

A total of 79,932 Chinese participants underwent routine physical examinations at Zhaobaoshan Residential District Community Health Service Center from Jan. 1, 2015 to Oct. 19, 2023.This study enrolled 6,882 individuals aged 60 years or older with DM who underwent liver ultrasonography in this cross-sectional study at Zhaobaoshan Residential District Community Health Service Center, from Jan. 1, 2015, to Oct. 19, 2023. And data was accessed for research purposes after Oct. 1,2024. Participants were randomly divided into a training group and a validation group in a 7:3 ratio. The diagnostic values of TyG and TyG-BMI were assessed using the area under the receiver-operating characteristic curve (AUROC) and Decision Curve Analysis (DCA). Two cut-off points were selected to rule out or rule in NAFLD, and we explored their specificity, sensitivity, negative predictive value, and positive predictive value.

Participants were excluded if they: (1) were younger than 60 years old; (2) had no history of diabetes or had FBG levels below 7.0 mmol/L; (3) had a history of liver disease (such as hepatitis B or C) or autoimmune liver disease; (4) had NAFLD combined with malignant tumors, or (5) had missing information on variables such as BMI, TG levels, FBG levels, or ultrasonic liver examination results. Ultimately, 6,882 individuals with T2DM aged 60 years or older who had undergone liver ultrasonography were enrolled in this cross-sectional study. The flowchart of the selection of participants is shown in Fig 1. All participants were randomly divided into a training set and a validation set at a ratio of 7:3.

Diabetes was defined as a FBG level >7.0mmol/L or a documented history of diabetes mellitus [15]. Hypertension was defined as a blood pressure reading of ≥140/90 mmHg or a history of hypertension [16]. NAFLD was diagnosed based on the presence of fatty liver, according to the guidelines outlined by Asia-Pacific Working Party [17]. Diagnosis was confirmed via ultrasound scans (using a Philips-Affiniti 30 machine) in all subjects after a minimum 8-hour fast. Presence

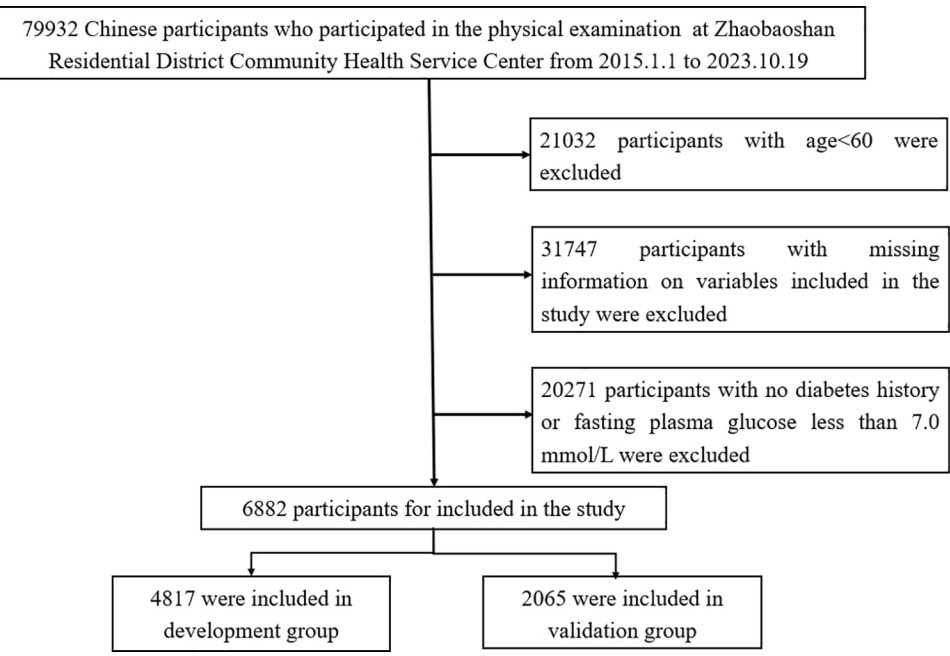

**Fig 1. The flowchart of the selection of participants in the study.**

of NAFLD was determined by increased echogenicity of the liver compared to the renal cortex and a fading rate of blood vessels. Participants were stratified into two groups based on the presence of NAFLD: the non-NAFLD group and the NAFLD group.

## 2.2. Demographic data and biochemical measurements

Baseline anthropometric measurements included age, gender, height, weight, heart rate, systolic blood pressure (SBP), and diastolic blood pressure (DBP). Blood samples were collected following a minimum 8-hour fast and analyzed for various biochemical parameters, including aspartate aminotransferase (AST), alanine aminotransferase (ALT), FBG, renal function markers (creatinine, urea nitrogen, and uric acid [UA]), and serum lipids (TG, total cholesterol [TC], high-density lipoprotein cholesterol [HDL-C], and low-density lipoprotein cholesterol [LDL-C]). All measurements were conducted using the ADVIA 2400 automatic biochemical analyzer (Siemens, Erlangen, Germany). Neutrophil count, lymphocyte count, hemoglobin, and platelet count were determined using the BC-6800 automatic five-part blood cell analyzer (Mindray, Shenzhen, China).

Height in a standing position and body weight without shoes were measured, and BMI was calculated as weight divided by the square of the height (kg/m^2). The TyG index was calculated using the formula: ln [(TG(mg/dL) × FBG (mg/dL))/2]. TyG-BMI = BMI × TyG.

## 2.3. Statistical analysis

All statistical analyses were conducted using R software, version 4.2.1 (R Foundation for Statistical Computing, Vienna, Austria). A *P value* of <0.05 was considered statistically significant. Results were presented as means with standard deviation (SD) for normally distributed continuous variables, median values with interquartile ranges (IQR) for non-normally distributed continuous variables, or frequencies and percentages for categorical variables. Missing data were treated as

deficiencies. Student's *t*-test for independent samples was employed for comparisons of normally distributed continuous variables, while the Mann-Whitney *U*-test was used for comparisons of non-normally distributed continuous variables. The Chi-square test was utilized for categorical variables.

To assess the robustness and independence of the associations between the exposure indices (TyG and TyG-BMI) and NAFLD across progressively more comprehensive sets of clinically relevant confounder, we identified and selected potential confounders, and subsequently constructed three hierarchical multivariable logistic regression models based on the results of univariate analyses in this study and the established evidence base. Odds ratios (ORs) with 95% confidence intervals (CIs) were calculated to estimate the risk of NAFLD. Variables that were statistically significant in univariate analyses were included in multivariate regression models [18]. Model 1 adjusted for fundamental demographic and basic vital sign parameters including age, systolic blood pressure (SBP), diastolic blood pressure (DBP), heart rate (HR), and self-reported history of hypertension. Model 2 (Model 1 + Hematological and inflammatory markers): Additionally adjusted for neutrophil count, lymphocyte count, hemoglobin (Hb), and platelet count (Plt) beyond Model 1. These hematological parameters were included because emerging evidence suggests that systemic inflammation and immune cell dysregulation play crucial roles in NAFLD pathogenesis and progression [19]. Neutrophil-to-lymphocyte ratio and platelet indices have been reported as independent predictors of NAFLD [19,20]. Model 3 (full model) further adjusted for liver function markers (ALT, AST), metabolic parameters (FBG, UA), and lipid profiles (TC, HDL-C, LDL-C) in addition to all variables in Model 2. This comprehensive model accounts for the direct metabolic and hepatic factors closely related to NAFLD pathophysiology [21–23]. Additionally, the Akaike Information Criterion (AIC) was employed to assess the relative fit of competing models. A lower AIC value indicates a model that better explains the data with greater parsimony, thereby guiding the selection of the most appropriate level of adjustment for our primary analysis. Variance Inflation Factors (VIFs) were calculated to assess multicollinearity among the independent variables in the model, and a VIF greater than 5 is commonly considered indicative of high multicollinearity. A *P* value for nonlinearity was determined by testing the null hypothesis (stating: second spline coefficient = 0). The RCS models were set for age, sex, blood pressure, heart rate, history of hypertension, lymphocytes, neutrophils, hemoglobin, platelets, AST and ALT levels, FBG, UA, TC, HDL-C, and LDL-C levels. Furthermore, to evaluate the dose-response relationships between TyG and TyG-BMI indices and NAFLD across different population characteristics, stratified analyses were performed by age groups (<40, 40–59, and ≥60 years) and hypertension status (presence or absence of hypertension history).

The predictive capabilities of TyG and TyG-BMI for diagnosing NAFLD were assessed using the AUROC curves. Comparisons of the AUROC values were conducted utilizing the Delong test. Specificity (SP), sensitivity (SE), negative predictive value (NPV), and positive predictive value (PPV) were calculated. The optimal cut-off value was established by maximizing the Youden index (sensitivity + specificity – 1) in both the training and validation groups. Furthermore, the net reclassification improvement (NRI) index and integrated discrimination improvement (IDI) index were computed to evaluate the incremental predictive value of TyG and TyG-BMI beyond established risk factors for NAFLD. Decision curve analysis (DCA) was performed to investigate the clinical utility of the predictive model with TyG and TyG-BMI in diagnosing NAFLD. The net benefit of decision-making was calculated by subtracting the proportion of false positives from the proportion of true positives, taking into account the relative risks of false-positive and false-negative outcomes. A *P* value of less than 0.05 was deemed statistically significant.

## 3. Results

### 3.1. Baseline characteristic of the study subjects

This study included 6,882 elderly subjects with diabetes mellitus, comprising 3,633 males and 3,249 females. Among them, 3,137 (45.58%) subjects had NAFLD, while 3,745 (54.41%) did not (Table 1). Apart from LDL-C, there were no significant differences in BMI, BP, HR, neutrophil count, lymphocyte count, Hb, Plt, liver function, FBG, kidney function, lipid panels, TyG, and TyG-BMI between the training and validation groups.

**Table 1. Baseline characteristics for the training and validation groups by NAFLD or non-NAFLD status in diabetes population.**

| Characteristics | Training Group | | | Validation Group | | |
|---|---|---|---|---|---|---|
| | Non-NAFLD | NAFLD | *P* value | Non-NAFLD | NAFLD | *P* value |
| N | 2607 | 2210 | | 1138 | 927 | |
| Age (mean (SD)) | 72.17 (7.52) | 70.31 (6.31) | <0.001 | 72.12 (7.53) | 70.58 (6.39) | <0.001 |
| Gender, Male (%) | 1351 (51.8) | 1193 (54.0) | 0.142 | 598 (52.5) | 491 (53.0) | 0.885 |
| BMI (Kg/m², mean (SD)) | 24.44 (49.63) | 25.97 (3.01) | 0.147 | 23.37 (2.94) | 26.03 (2.94) | <0.001 |
| SBP (mmHg, mean (SD)) | 139.85 (19.01) | 150.16 (233.66) | 0.025 | 138.93 (17.86) | 144.38 (17.94) | <0.001 |
| DBP (mmHg, mean (SD)) | 77.32 (9.45) | 79.72 (9.62) | <0.001 | 76.80 (9.37) | 80.18 (9.74) | <0.001 |
| HR (bpm, mean (SD)) | 73.20 (11.46) | 74.21 (11.94) | 0.003 | 73.17 (11.66) | 74.40 (12.00) | 0.019 |
| Hypertension, Yes (%) | 2065 (79.2) | 1868 (84.5) | <0.001 | 879 (77.2) | 787 (84.9) | <0.001 |
| Neutrophil (10^9/L, mean (SD)) | 3.67 (1.22) | 3.85 (1.23) | <0.001 | 3.66 (1.24) | 3.86 (1.21) | <0.001 |
| Lymphocyte (10^9/L, mean (SD)) | 1.92 (0.62) | 2.18 (0.65) | <0.001 | 1.94 (0.64) | 2.19 (0.65) | <0.001 |
| Hb (g/L,mean (SD)) | 137.00 (15.10) | 141.52 (13.97) | <0.001 | 136.28 (14.83) | 142.08 (13.95) | <0.001 |
| Plt (10^12/L, mean (SD)) | 197.98 (57.01) | 209.30 (53.50) | <0.001 | 197.54 (59.43) | 208.60 (54.80) | <0.001 |
| ALT (U/L, mean (SD)) | 21.37 (25.22) | 28.03 (19.38) | <0.001 | 20.82 (13.36) | 27.89 (17.02) | <0.001 |
| AST (U/L, mean (SD)) | 26.54 (27.22) | 29.88 (15.24) | <0.001 | 26.17 (11.30) | 29.98 (15.95) | <0.001 |
| FBG (mmol/L, mean (SD)) | 7.06 (2.22) | 7.40 (2.12) | <0.001 | 7.04 (2.16) | 7.37 (1.97) | <0.001 |
| UA (µmol/L, mean (SD)) | 335.24 (92.04) | 351.98 (89.32) | <0.001 | 333.00 (90.40) | 351.33 (87.80) | <0.001 |
| Urea nitrogen (mmol/L, mean (SD)) | 5.66 (1.93) | 5.33 (1.58) | <0.001 | 5.62 (1.78) | 5.44 (1.62) | 0.021 |
| Creatinine (µmol/L, mean (SD)) | 78.60 (25.69) | 75.80 (19.38) | <0.001 | 77.97 (29.06) | 77.48 (24.20) | 0.68 |
| Triglyceride (mmol/L, mean (SD)) | 1.48 (0.92) | 2.07 (1.30) | <0.001 | 1.46 (0.94) | 2.08 (1.41) | <0.001 |
| Total Cholestrol (mmol/L, mean (SD)) | 4.62 (1.21) | 4.62 (1.23) | 0.915 | 4.63 (1.22) | 4.69 (1.31) | 0.29 |
| HDL-C (mmol/L, mean (SD)) | 1.69 (0.52) | 1.61 (0.46) | <0.001 | 1.71 (0.50) | 1.62 (0.43) | <0.001 |
| LDL-C (mmol/L, mean (SD)) | 2.59 (0.94) | 2.73 (0.96) | <0.001 | 2.62 (0.96) | 2.81 (1.01) | <0.001 |
| TyG (mean (SD)) | 8.85 (0.58) | 9.24 (0.59) | <0.001 | 8.84 (0.57) | 9.24 (0.61) | <0.001 |
| TyG-BMI (mean (SD)) | 216.89 (445.01) | 240.07 (32.11) | 0.015 | 206.96 (31.56) | 240.48 (31.82) | <0.001 |

SD, standard deviation; BMI, body mass index; SBP, systolic blood pressure; DBP, diastolic blood pressure; HR, heart rate; Hb, hemoglobin; Plt, platelet count; ALT, alanine aminotransferase; AST, aspartate aminotransferase; FBG, fasting blood glucose; UA, uric acid; HDL-C, high-density lipoprotein cholesterol; LDL-C, low-density lipoprotein cholesterol; TyG, triglyceride-glucose; TyG-BMI, triglyceride-glucose-body mass index.

The subjects were randomly allocated into the training group (n = 4,817) and the validation group (n = 2,065) (S1 Table). In the training group, 2,210 subjects (45.87%) were diagnosed with NAFLD, while in the validation group, 927 subjects (44.89%) had NAFLD (Table 1). When stratified by age into 10 intervals, the prevalence of NAFLD decreased gradually in both male and female subjects as age increased. The lowest prevalence was observed in subjects older than 80, with rates of 28.79% and 32.47% in the training and validation groups, respectively (S2 Table).

Among subjects with NAFLD in the training group, the levels of TyG and TyG-BMI were 9.24 ± 0.59 and 240.07 ± 32.11, respectively, which were significantly higher than those of subjects without NAFLD. In the training group, elderly diabetic subjects with NAFLD exhibited significantly elevated levels of SBP, DBP, HR, Hb, Plt, ALT, AST, FBG, triglycerides, LDL and UA compared to non-NAFLD subjects. Conversely, non-NAFLD subjects had significantly higher levels of HDL, urea nitrogen, and creatinine compared to NAFLD subjects (Table 1). Except for BMI and serum creatinine, the comparison of other indexes between non-NAFLD and NAFLD subjects in the validation group was consistent with that in the training group.

 

### 3.2. Association of TyG and TyG-BMI with NAFLD in elderly participants with diabetes

The indicators showing significant differences between non-NAFLD and NAFLD subjects were included in multivariate regression analysis. In the minimally adjusted model, an increase in TyG and TyG-BMI was associated with an increased risk of NAFLD for every unit raised (OR=2.959, 95%CI: 2.650–3.304, P < 0.001; OR=1.032, 95%CI: 1.029–1.034, P < 0.001; respectively). Similarly, in the fully adjusted model, after controlling for confounders, TyG and TyG-BMI remained correlated with an increased risk of NAFLD for every unit raised (OR=3.920, 95%CI: 3.358–4.576, P < 0.001; OR=1.032, 95%CI: 1.029–1.034, P < 0.001; respectively). These findings were consistent in the validation group, both in the minimally and fully adjusted models (Table 2). Otherwise, the AIC values exhibited a sequential decline across Models 1, 2, and 3 in both the training and validation group, which supports the rationale for including additional covariates. (Table 2). A VIF value of less than 5 for each variable in the models suggests that multicollinearity is not a concern.

To elucidate potential effect modification and visualize dose-response relationships within key subgroups (age and hypertension status), TyG and TyG-BMI were analyzed both continuously and by quartiles in subgroup models for the overall training dataset. For these stratified analyses, we prioritized model parsimony and visualization clarity by adjusting only for those covariates that demonstrated statistically significant differences (P < 0.05) between NAFLD and non-NAFLD participants. The results demonstrated that in each subgroup model, after adjustment for covariates, both TyG and TyG-BMI indices remained significantly and positively associated with NAFLD. In age subgroups, the ORs for NAFLD were highest in the highest quartile group of baseline TyG-BMI compared to the lowest quartile group, across age-stratified intervals. Similarly, in the hypertension subgroup, the ORs for NAFLD were highest in the highest quartile group compared to the other quartile groups (Table 3A). The ORs for NAFLD associated with TyG were also consistent with those for TyG-BMI (Table 3B).

As depicted in Fig 2, there was a non-linear relationship between TyG and TyG-BMI and the prevalence of NAFLD (*P* for nonlinearity<0.01). The RCS curve in Fig 2 indicated that at TyG-BMI = 220.5952 and TyG = 8.984(reference) in the training group, a higher level of TyG-BMI (a) or TyG (b) was associated with an increased risk of NAFLD, as well as those of TyG-BMI (c) or TyG(d) in the validation.

**Table 2. Multivariate analysis for the association of triglyceride-glucose-body mass index and triglyceride-glucose index with fatty liver disease in training and validation groups.**

|  | Model | Training Group | | | Validation Group | | |
|---|---|---|---|---|---|---|---|
|  |  | OR (95%CI) | *P* value | AIC | OR (95%CI) | *P* value | AIC |
| TyG-BMI | Model 1 | 1.032(1.029-1.034) | <0.001 | 6052.3 | 1.030(1.026-1.033) | <0.001 | 2308.3 |
|  | Model 2 | 1.031(1.028-1.033) | <0.001 | 5978.2 | 1.028(1.025-1.032) | <0.001 | 2276.5 |
|  | Model 3 | 1.032(1.029-1.034) | <0.001 | 5953.0 | 1.030(1.026-1.034) | <0.001 | 2244.0 |
| TyG | Model 1 | 2.959(2.650-3.304) | <0.001 | 6034.7 | 3.092(2.609-3.664) | <0.001 | 2557.6 |
|  | Model 2 | 2.686(2.401-3.004) | <0.001 | 5928.5 | 2.722(2.288-3.238) | <0.001 | 2502.3 |
|  | Model 3 | 3.920(3.358-4.576) | <0.001 | 5792.7 | 3.943(3.119-4.984) | <0.001 | 2439.9 |

Model 1: age, blood pressure and heart rate, the history of hypertension.

Model 2: age, blood pressure, heart rate, the history of hypertension, lymphocytes, neutrophils, hemoglobin and platelets.

Model 3: age, blood pressure, heart rate, the history of hypertension, lymphocytes, neutrophils, hemoglobin, platelets, AST and ALT levels, FBG, UA, TC, and HDL-C, LDL-C levels.

OR, odds ratio; TyG-BMI, triglyceride-glucose-body mass index; TyG, triglyceride-glucose; AST, aspartate aminotransferase; ALT, alanine aminotransferase; FBG, fasting blood glucose; UA, uric acid; TC, total cholesterol; HDL-C, high-density lipoprotein cholesterol; LDL-C, low-density lipoprotein cholesterol; AIC, Akaike Information Criterion.

**Table 3. The association of triglyceride-glucose-body mass index (A) and triglyceride-glucose (B) index with fatty liver disease in training dataset by age group and self-report hypertension.**

**A**

| Characteristic | Continuous | | Quantile 1(<197.398) | Quantile 2(197.398–221.085) | | Quantile 3(221.085–246.099) | | Quantile 4(>246.099) | | P for trend |
|---|---|---|---|---|---|---|---|---|---|---|
| | OR (95%CI) | P value | | OR (95%CI) | P value | OR(95%CI) | P value | OR(95%CI) | P value | |
| Age group | | | | | | | | | | |
| 60-69 | 1.030 (1.025-1.035) | <0.001 | Ref | 4.418 (3.180-6.138) | <0.001 | 9.926 (7.099-13.879) | <0.001 | 16.863 (11.775-24.151) | <0.001 | <0.001 |
| 70-79 | 1.032 (1.025-1.038) | <0.001 | Ref | 2.728 (2.009-3.704) | <0.001 | 5.783 (4.219-7.928) | <0.001 | 14.685 (10.375-20.785) | <0.001 | <0.001 |
| >80 | 1.027 (1.017-1.037) | <0.001 | Ref | 2.990 (1.465-6.104) | <0.001 | 10.543(5.202-21.367) | <0.001 | 14.374 (6.807-30.353) | <0.001 | <0.001 |
| Self-report hypertension | | | | | | | | | | |
| No | 1.028 (1.024-1.031) | <0.001 | Ref | 4.701 (2.849-7.757) | <0.001 | 12.765 (7.599-21.441) | <0.001 | 20.946 (11.597-37.831) | <0.001 | <0.001 |
| Yes | 1.040 (1.022-1.059) | <0.001 | Ref | 2.908 (2.301-3.674) | <0.001 | 6.628 (5.228-8.403) | <0.001 | 13.284 (10.311-17.114) | <0.001 | <0.001 |

**B**

| Characteristic | Continuous | | Quantile 1(<8.612) | Quantile 2(8.612–8.982) | | Quantile 3(8.982–9.398) | | Quantile 4(>9.398) | | P for trend |
|---|---|---|---|---|---|---|---|---|---|---|
| | OR (95%CI) | P value | | OR(95%CI) | P value | OR(95%CI) | P value | OR(95%CI) | P value | |
| Age group | | | | | | | | | | |
| 60-69 | 3.152 (2.402-4.136) | <0.001 | Ref | 2.541 (1.899-3.400) | <0.001 | 3.647 (2.680-4.963) | <0.001 | 7.187 (5.030-10.270) | <0.001 | <0.001 |
| 70-79 | 3.023 (1.984-4.607) | <0.001 | Ref | 2.748 (2.058-3.668) | <0.001 | 3.912 (2.878-5.318) | <0.001 | 6.199 (4.265-9.012) | <0.001 | <0.001 |
| >80 | 2.035 (1.244-3.329) | <0.001 | Ref | 3.308 (1.740-6.291) | <0.001 | 5.988 (3.126-11.469) | <0.001 | 8.727 (4.175-18.242) | <0.001 | <0.001 |
| Self-report hypertension | | | | | | | | | | |
| No | 3.456 (2.011-5.941) | <0.001 | Ref | 2.236 (1.705-2.932) | <0.001 | 3.085 (2.304-4.129) | <0.001 | 6.701 (4.699-9.554) | <0.001 | <0.001 |
| Yes | 2.440 (1.988-2.993) | <0.001 | Ref | 2.893 (2.180-3.838) | <0.001 | 4.061 (3.029-5.444) | <0.001 | 6.012 (4.297-8.411) | <0.001 | <0.001 |

OR, odds ratio; CI, odds ratio; Ref, Reference. Adjusting variables: blood pressure, heart rate, lymphocytes, neutrophils, hemoglobin, platelets, AST and ALT levels, FBG, UA, TC, and HDL-C, LDL-C levels.

### 3.3. Comparison of TyG and TyG-BMI as predictive indicators for NAFLD

ROC curve analysis was conducted with NAFLD as the state variable and baseline TyG and TyG-BMI as test variables in both the training and validation groups. The AUCs of both indicators indicated their predictive efficacy for NAFLD in both groups, with TyG-BMI exhibiting the highest predictive accuracy (Fig 3). A comparison of AUCs revealed that TyG-BMI had a significantly higher AUC than TyG ($P < 0.05$).

In the training group, the optimal cut-off for TyG was >8.929, achieving 69.0% sensitivity and 58.5% specificity, whereas for TyG-BMI, a cut-off of >219.028 yielded 74.8% sensitivity and 65.7% specificity. Additionally, the PPV for TyG-BMI (0.649) was higher compared to TyG (0.585), while the NPV for TyG-BMI (0.754) was higher compared to TyG (0.690). Similarly, in the validation group, both the PPV and the NPV for TyG-BMI were higher than TyG (Table 4).

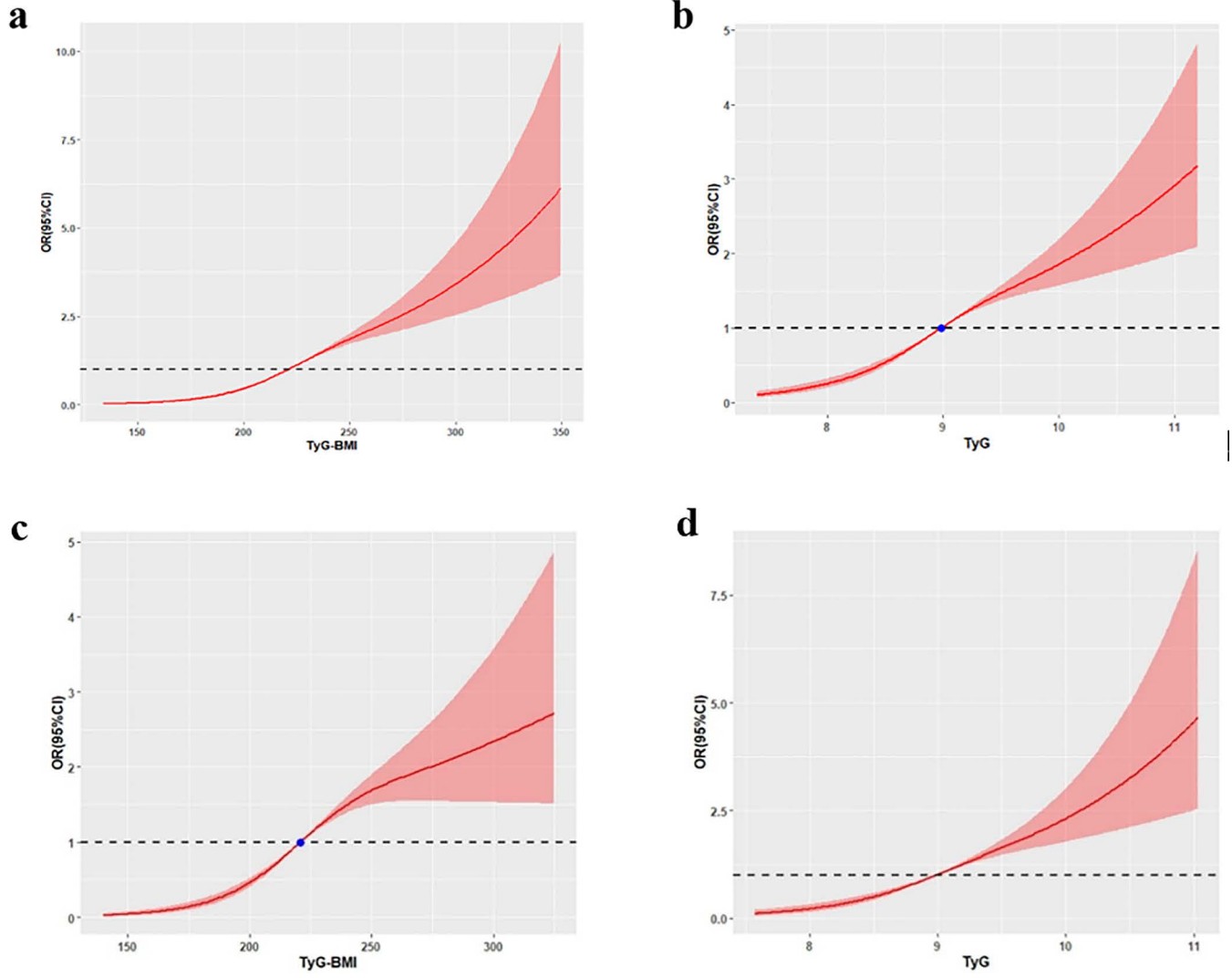

**Fig 2. Non-linear relationship between TyG and TyG-BMI and the prevalence of NAFLD. (a)(b)** Restricted cubic spline curves for the TyG-BMI and TyG in the training group. **(c)(d)** Restricted cubic spline curves for the TyG-BMI and TyG in the validation group.

Predictive ability and reclassification statistics of the baseline model with and without the TyG-BMI index and TyG index were also evaluated in all participants. Compared to the baseline model without TyG-BMI, the continuous IDI reached 0.5593 (95%CI 0.5048–0.6138), with 0.0125 (95%CI 0.0114–0.0137) of NRI when the TyG-BMI index was included. TyG-BMI demonstrated a significant improvement in both NRI and IDI. Additionally, the continuous IDI of TyG-BMI was higher than TyG. Overall, TyG-BMI exhibited superior predictive efficacy for NAFLD compared to TyG (Table 5).

### 3.4. The diagnostic accuracy of TyG-BMI for NAFLD in elderly participants with diabetes

As previously mentioned, TyG-BMI showed superior predictive value for NAFLD compared to TyG. Herein, we focus on analyzing the diagnostic accuracy of TyG-BMI for NAFLD in elderly participants with diabetes. Table 6 presented the diagnostic accuracy of TyG-BMI in predicting NAFLD at decile intervals. In the training group, when the cut-off point

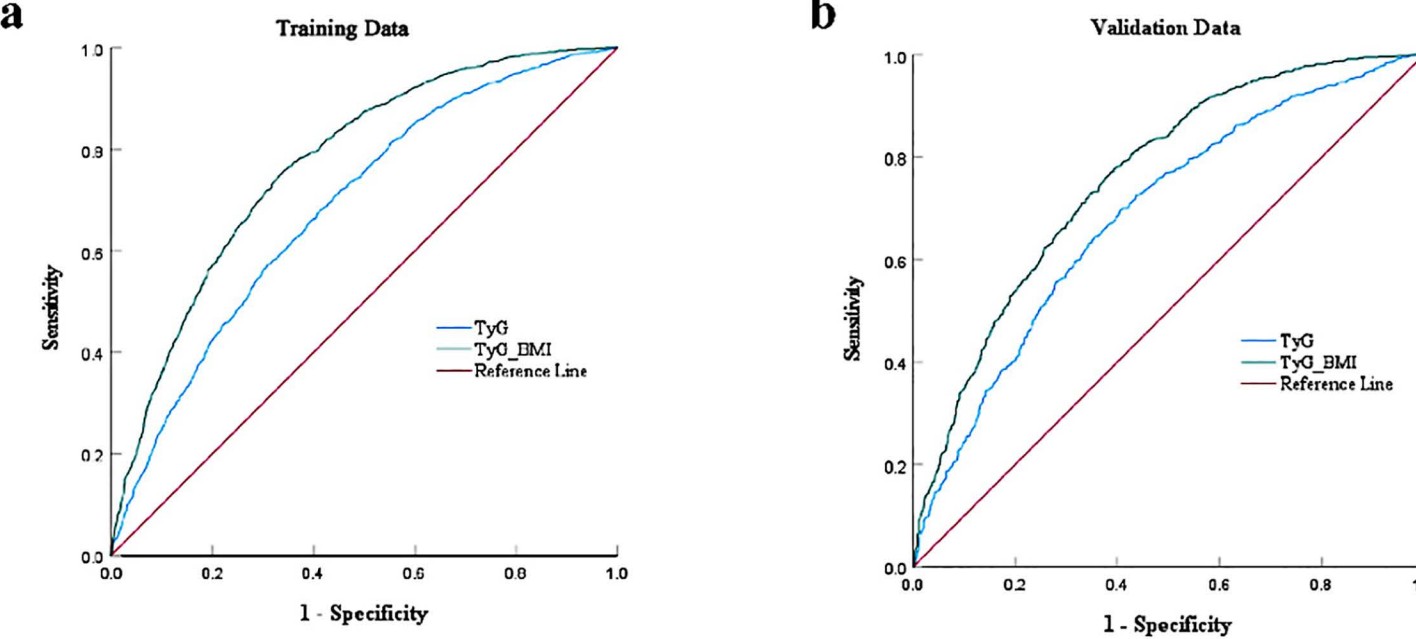

**Fig 3. Receiver operating characteristic (ROC) curves of the TyG-BMI and TyG for diagnosing NAFLD. (a)** The ROC of TyG-BMI and TyG in the training group. **(b)** The ROC of TyG-BMI and TyG in the validation group.

**Table 4. The optimal cut-off point for the triglyceride-glucose-body mass index and triglyceride-glucose in diagnosing non-alcoholic fatty liver disease in training and validation group.**

| Dataset | Marker | Cut-off | AUC (95%CI) | P value* | Sensitivity | Specificity | PPV | NPV |
|---|---|---|---|---|---|---|---|---|
| Training | TyG-BMI | 219.028 | 0.771(0.758-0.784) | Reference | 0.748 | 0.657 | 0.649 | 0.754 |
| | TyG | 8.929 | 0.678(0.663-0.692) | <0.001 | 0.690 | 0.585 | 0.585 | 0.690 |
| Validation | TyG-BMI | 215.712 | 0.756(0.738-0.779) | Reference | 0.770 | 0.626 | 0.641 | 0.759 |
| | TyG | 9.109 | 0.703(0.680-0.725) | <0.001 | 0.586 | 0.723 | 0.646 | 0.668 |

AUC, area under curve; PPV, positive predictive value; NPV, negative predictive value; TyG-BMI, triglyceride-glucose-body mass index; TyG, triglyceride-glucose.

* P value for DeLong test

**Table 5. Added predictive ability and reclassification statistics of triglyceride-glucose-body mass index and triglyceride-glucose index.**

| Model | Continuous NRI (95%CI) | P value | IDI (95%CI) | P value |
|---|---|---|---|---|
| Baseline Model | Ref | | Ref | |
| Baseline Model+TyG-BMI index | 0.5593 (0.5048 - 0.6138) | <0.001 | 0.0125 (0.0114 - 0.0137) | <0.001 |
| Baseline Model | Ref | | Ref | |
| Baseline Model+TyG index | 0.3718 (0.3161 - 0.4275) | <0.001 | 0.0407(0.035 - 0.0465) | <0.001 |

NRI, net reclassification improvement; CI, integrated discrimination improvement; IDI, integrated discrimination improvement; TyG-BMI, triglyceride-glucose-body mass index; TyG, triglyceride-glucose; Ref, Reference. Baseline Model: including the covariates from the full model (Model 3).

of TyG-BMI was set at 212.886 to discriminate NAFLD, it exhibited a relatively high Youden's index (0.381) and diagnostic accuracy of sensitivity (0.806)/specificity(0.575), PPV(0.617)/NPV(0.778). Conversely, when the cut-off point of TyG-BMI was set at 251.741, the diagnostic accuracy of sensitivity (0.326)/specificity (0.907), PPV (0.748)/NPV (0.614) was observed. Thus, a TyG-BMI < 212.886 could be utilized to rule out NAFLD (SE = 80.6%, NPV = 77.8%), while a TyG-BMI ≥ 251.741 could be used to rule in NAFLD (SP = 90.7%, PPV = 74.8%) (Table 6).

In the validation group, when the cut-off point of TyG-BMI was set at 212.886 to discriminate NAFLD, it achieved the highest Youden's index (0.389), with diagnostic accuracy of sensitivity (0.800)/specificity (0.590), PPV(0.628)/NPV(0.773). Conversely, when the cut-off point of TyG-BMI was set at 251.741, the diagnostic accuracy of sensitivity (0.318)/specificity (0.915), PPV (0.764)/NPV (0.608) was observed. Therefore, a TyG-BMI < 212.886 could be utilized to rule out NAFLD (SE = 80.0%, NPV = 77.3%), while a TyG-BMI ≥ 251.741 could be used to rule in NAFLD (SP = 91.5%, PPV = 76.4%) (Table 6). These two cut-off points may thus be valuable in diagnosing NAFLD in elderly participants with diabetes (Table 6).

To contextualize these findings within existing clinical frameworks, comparisons were made with prior studies. For example, Li *et al.* (2024) reported a TyG-BMI cut-off of 229 for identifying hyperuricemia among patients with non-alcoholic fatty liver disease (NAFLD-HUA), achieving an AUC of 0.656 and outperforming the TyG index alone (AUC = 0.605, P = 0.001) [(34)]. Although the clinical outcome in their study differed from ours, their cut-off value falls within a similar range. Furthermore, their quartile stratification (Q1 ≤ 218.46; Q4 > 262.21) closely aligns with our ROC-derived thresholds. These consistencies suggest that TyG-BMI values between approximately 220 and 260 may have generalizable diagnostic relevance across diverse metabolic diseases.

### 3.5. Clinical application of the model

Fig 4 illustrated the DCA of the predictive models with TyG and TyG-BMI in both the training (Fig 4a) and validation groups (Fig 4b). In these figures, the black line represented the net benefit when no participants were diagnosed with NAFLD,

**Table 6. Diagnostic accuracy of the triglyceride-glucose-body mass index at decile intervals.**

|  | Cut-off | Sensitivity | Specificity | PPV | NPV | Youden Index |
|---|---|---|---|---|---|---|
| Training | 177.748 | 0.986 | 0.173 | 0.503 | 0.936 | 0.159 |
|  | 191.811 | 0.948 | 0.326 | 0.544 | 0.882 | 0.274 |
|  | 202.890 | 0.887 | 0.459 | 0.582 | 0.828 | 0.346 |
|  | 212.886 | 0.806 | 0.575 | 0.617 | 0.778 | 0.381 |
|  | 221.489 | 0.718 | 0.684 | 0.658 | 0.741 | 0.402 |
|  | 230.529 | 0.600 | 0.769 | 0.688 | 0.694 | 0.369 |
|  | 240.570 | 0.468 | 0.843 | 0.716 | 0.652 | 0.311 |
|  | 251.741 | 0.326 | 0.907 | 0.748 | 0.614 | 0.233 |
|  | 269.065 | 0.169 | 0.959 | 0.776 | 0.576 | 0.128 |
| Validation | 177.748 | 0.987 | 0.161 | 0.505 | 0.937 | 0.148 |
|  | 191.811 | 0.948 | 0.322 | 0.548 | 0.877 | 0.270 |
|  | 202.890 | 0.883 | 0.463 | 0.587 | 0.821 | 0.346 |
|  | 212.886 | 0.800 | 0.590 | 0.628 | 0.773 | 0.389 |
|  | 221.489 | 0.697 | 0.689 | 0.660 | 0.725 | 0.387 |
|  | 230.529 | 0.570 | 0.768 | 0.680 | 0.674 | 0.338 |
|  | 240.570 | 0.442 | 0.849 | 0.717 | 0.637 | 0.291 |
|  | 251.741 | 0.318 | 0.915 | 0.764 | 0.608 | 0.233 |
|  | 269.065 | 0.172 | 0.958 | 0.782 | 0.572 | 0.131 |

PPV, positive predictive value; NPV, negative predictive value.

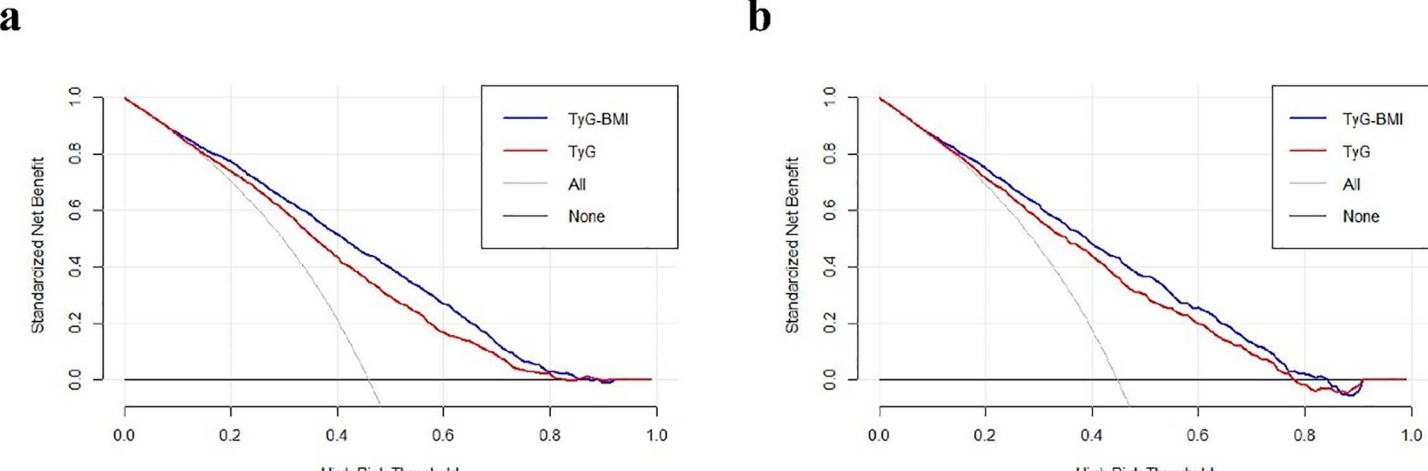

**Fig 4. Decision Curve of the predictive model with TyG-BMI or TyG for NAFLD in the training and validation group. (a)** The DCA of **the predictive models with** TyG and TyG-BMI in the training groups. **(b)** The **the predictive models with** of TyG and TyG-BMI in the validation groups.

while the light gray line represented the net benefit when everyone was diagnosed with NAFLD. The diagnostic utility of a model was determined by the distance between the "no treatment line" (black line) and the "all treatment line" (light gray line) in its curve. In clinical practice, a model is considered more useful when its curve is further from both the black and light gray lines. Hence, the clinical application of TyG-BMI is preferable when its curve is significantly distant from both lines, whether in the training or validation group.

## 4. Discussion

Non-alcoholic fatty liver disease (NAFLD) poses a substantial burden in elderly individuals with diabetes mellitus, necessitating simple, non-invasive screening strategies for early detection and risk stratification. In this cross-sectional study of 6,882 elderly diabetic patients, we evaluated the diagnostic utility of the triglyceride-glucose index (TyG) and triglyceride-glucose-body mass index (TyG-BMI) for NAFLD identification. Both TyG and TyG-BMI were significantly elevated in NAFLD patients across training and validation cohorts, with these associations persisting after comprehensive adjustment covariates. Notably, TyG-BMI demonstrated superior predictive accuracy compared to TyG, with greater clinical net benefit confirmed by decision curve analysis. We identified two clinically actionable cut-off points: TyG-BMI < 212.886 effectively ruled out NAFLD (sensitivity ~80%, NPV ~77%), while TyG-BMI ≥ 251.741 ruled in NAFLD (specificity ~91%, PPV ~75%) in both groups. These findings establish TyG-BMI as a practical, cost-effective screening tool for NAFLD in elderly diabetic populations using routinely available parameters.

In this study, the prevalence of NAFLD among elderly individuals with diabetes was 45.58%, notably higher than that reported in the general population [4]. However, this prevalence remains lower than the approximately 75% reported among patients with T2DM in broader studies [10]. Interestingly, we observed a stepwise decline in NAFLD prevalence with increasing age among both elderly males and females. This trend is consistent with findings from studies in Japanese populations, which report a progressive decrease in NAFLD prevalence among males over 50 and females over 60, compared to their younger counterparts [4]. This finding can be attributed to several age-related physiological changes. Specifically, sarcopenia (age-related loss of muscle mass), unintentional weight loss, and altered adipose tissue distribution and function in the very elderly population may collectively reduce hepatic fat accumulation [24,25]. Additionally, survivor bias may play a role, as individuals with severe NAFLD and its complications may have higher mortality rates, leading to a healthier survivor cohort in the oldest age groups.

A variety of non-invasive and user-friendly models have been developed to facilitate the detection and evaluation of NAFLD [26,27]. Previous studies have demonstrated that obesity-related and lipid-related indices, such as the lipid accumulation product (LAP), hepatic steatosis index (HSI), and visceral adiposity index (VAI), are effective predictors of NAFLD [28–30]. However, these indices were mainly developed based on data from Western populations and may have limited applicability in Asian populations. Emerging evidence indicates that indices derived from the TyG index, which are considered more appropriate for Asian populations, can effectively reflect insulin resistance and predict the presence of NAFLD. The etiology of NAFLD is multifactorial, involving a complex interaction among environmental exposures, genetic predisposition, and dietary habits [31]. Dietary patterns, in particular, play a significant role in the progression of NAFLD, and elderly individuals often exhibit distinct dietary behaviors compared to younger populations. Nevertheless, it remains uncertain whether parameters related to the TyG index can reliably predict NAFLD in elderly individuals.

In this cross-sectional study, parameters associated with the TyG index demonstrated independent correlations with NAFLD in elderly patients diagnosed with diabetes mellitus. Furthermore, elevated TyG and TyG-BMI indexes were linked to an increased risk of developing NAFLD. The TyG index, in conjunction with both TyG and FBG levels, has been proposed as an effective surrogate marker for IR. Recent research has highlighted the role of adipose tissue in the development of IR, potentially through the release of lipids and various circulating factors that can exacerbate IR in other organs. Additionally, excessive triglyceride accumulation in the liver may further contribute to the induction of IR.

We compared the predictive value of the TyG-BMI index and the TyG index for identifying NAFLD in elderly patients with T2DM in the present study. The TyG-BMI index, originally proposed by Wang *et al.* in a Japanese cohort, has been validated as an effective tool for predicting NAFLD [18]. Consistent with previous findings, our results demonstrate that TyG-BMI exhibits superior predictive accuracy compared to the TyG index in this specific population. Prior studies have indicated that TyG-BMI has greater predictive utility for NAFLD in individuals with obesity than in those without obesity [9,32]. Moreover, Lim *et al.* reported that TyG-BMI outperforms the TyG index and other TyG-related parameters in the assessment of insulin resistance [33]. Eguchi *et al.* further emphasized that NAFLD prevalence increases linearly with increasing BMI and that obesity is an independent risk factor for NAFLD, regardless of other metabolic abnormalities [34]. Interestingly, other research has shown a stronger association between the TyG index and NAFLD risk in non-obese individuals compared to those with obesity [35]. These findings suggest that BMI may modify the association between TyG-related parameters and NAFLD risk. This interaction may account for the superior performance of TyG-BMI over the TyG index in predicting NAFLD, particularly in elderly patients with diabetes.

In this study, we investigated two cut-off values of TyG-BMI for identifying or excluding NAFLD, along with their corresponding positive and negative predictive values in both the training and validation groups. The TyG-BMI, a simple, non-invasive, and cost-effective tool, accurately distinguished participants with and without NAFLD. In the training group, the cut-off values were 212.886 for a sensitivity of 0.806 and 251.741 for a specificity of 0.907, resulting in a negative predictive value of 0.778 and a positive predictive value of 0.748. When applying this model to the validation group, a TyG-BMI < 212.886 could be used to rule out NAFLD (SE = 80.0%, NPV = 77.3%), while a TyG-BMI ≥ 251.741 could be used to rule in NAFLD (SP = 91.5%, PPV = 76.4%). Therefore, the TyG-BMI model demonstrated promising clinical utility. Hu *et al.* previously demonstrated in a Japanese cohort that two TyG-BMI cut-off values (182.2 and 224.0) could accurately identify NAFLD and reduce unnecessary ultrasonography in the population [4]. Xuejie Wang *et al.* further determined the optimal TyG-BMI cut-off value of 218.08 in a Chinese general population [36]. Notably, while the cut-off values in our study were higher than those reported in the Japanese population, they showed similarity to the thresholds established in Chinese populations. We speculate that differences in dietary habits between elderly patients and the general population may contribute to this variation. The Japanese diet, rich in essential fatty acids, may reduce hepatic inflammation, fibrosis, and steatosis [37]. Moreover, Chinese dietary patterns differ from those in Japan, and blood glucose levels in our study were higher than in the general population. Therefore, constructing different ranges of TyG-BMI to predict NAFLD in specific populations may be warranted.

The liver plays a central role in the regulation of glucose and lipid metabolism. A hallmark of NAFLD is the excessive accumulation of TG, primarily driven by hepatic *de novo* lipogenesis (DNL). Within hepatocytes, endogenously synthesized TGs are efficiently assembled into VLDL particles containing apoB100, which facilitate TG transport through the bloodstream to peripheral tissues such as muscle and adipose tissue [31,38]. In the setting of insulin resistance, hyperinsulinemia robustly promotes DNL via activation of the transcription factor SREBP-1c. Concurrently, elevated glucose levels activate the ChREBP, further enhancing lipogenic activity [18,34]. NAFLD develops when the hepatic influx of circulating fatty acids and TGs, in conjunction with increased DNL, exceeds the liver's capacity for mitochondrial β-oxidation and VLDL-TG secretion [4].

The TyG-BMI index has emerged as a promising tool for NAFLD screening due to its simplicity, non-invasiveness, and reliance on routine clinical parameters including BMI, fasting blood glucose, and triglyceride levels. It offers a cost-effective alternative to imaging modalities, particularly in resource-limited primary care settings. In this study, a TyG-BMI cut-off value of ≥251.741 yielded high specificity (91.5%) and a positive predictive value (76.4%), making it suitable for identifying high-risk individuals who may benefit from further diagnostic imaging. Conversely, a cut-off value <212.886 demonstrated high sensitivity (80.0%) and a negative predictive value (77.3%), suggesting its utility in ruling out NAFLD. These thresholds were derived from a cohort of elderly Chinese patients with diabetes; however, given the potential influence of ethnicity and body composition, further validation in diverse populations is warranted. Despite this, the consistency of findings across multiple studies supports the broader applicability of the TyG-BMI index. Integration of this index into screening algorithms and electronic health record (EHR) systems may improve early detection and disease monitoring, particularly among elderly diabetic individuals who are often underdiagnosed.

This study has several limitations. First, due to its cross-sectional design, causal inferences cannot be established. Second, the lack of liver biopsy, which remains the gold standard for diagnosing and staging NAFLD, represents a notable limitation. While ultrasonography is widely used for NAFLD detection in clinical practice, its limited sensitivity, particularly in cases with mild steatosis, may lead to underdiagnosis. This may introduce selection or information bias and affect the study's accuracy and reliability. Third, external validation was not performed, which limits the generalizability of the proposed model to other populations. Fourth, due to the length limitation and research direction of the paper, there is no further analysis on the low incidence rate of NAFLD among the elderly. We will analyze the impact of age on NAFLD in the elderly in future research. Additionally, although our models were adjusted for several relevant covariates, including liver enzymes such as AST and ALT, other important metabolic markers such as HbA1c and insulin levels were not included due to incomplete data. These missing variables may serve as residual confounders and should be addressed in future research.

## 5. Conclusions

In conclusion, although the prevalence of NAFLD remained high among elderly individuals with diabetes, a stepwise decline was observed with increasing age in both elderly males and females. In this population, the TyG-BMI index demonstrated superior predictive accuracy for NAFLD compared to the TyG index alone. Given its simplicity, non-invasiveness, and low cost, the TyG-BMI index holds particular promise for implementation in primary care settings as a practical screening tool to identify elderly diabetic patients at high risk for NAFLD, thereby underscoring its clinical utility in early detection and prevention strategies.

## Supporting information

**S1 Table. The baseline characteristics for the training and validation groups in diabetes population.**
(DOCX)

**S2 Table. The prevalence of NAFLD in different age groups in the training and validation groups.**
(DOCX)

 

# Acknowledgement

We thanked the participants and staff at Zhaobaoshan Residential District Community Health Service Center, Zhenhai, Ningbo.

# Author contributions

**Conceptualization:** Gaohui Zhu, Kedan Cai.

**Data curation:** Gaohui Zhu, Yihui Qu, Dingfa He, Ziwei Tang, Xinyi Wang, Minqiao Zhang, Peilan Jiang.

**Formal analysis:** Jing Wang, Ruijie Zhang.

**Writing – original draft:** Gaohui Zhu, Kanan Chen.

**Writing – review & editing:** Kedan Cai.

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
