## [Decision Letter · Decision Letter 0]

7 May 2025

Dear Dr. Cai,

Thank you for submitting your manuscript to PLOS ONE. After careful consideration, we feel that it has merit but does not fully meet PLOS ONE’s publication criteria as it currently stands. Therefore, we invite you to submit a revised version of the manuscript that addresses the points raised during the review process.

Please make peer-to-peer modifications to the reviewer's comments.

We look forward to receiving your revised manuscript.

Kind regards,

Qian Wu

Academic Editor

PLOS ONE

Reviewers' comments:

Reviewer's Responses to Questions

**Comments to the Author**

1. Is the manuscript technically sound, and do the data support the conclusions?

Reviewer #1: Yes

Reviewer #2: Yes

Reviewer #3: Yes

2. Has the statistical analysis been performed appropriately and rigorously?

Reviewer #1: Yes

Reviewer #2: Yes

Reviewer #3: Yes

3. Have the authors made all data underlying the findings in their manuscript fully available?

Reviewer #1: Yes

Reviewer #2: Yes

Reviewer #3: Yes

4. Is the manuscript presented in an intelligible fashion and written in standard English?

Reviewer #1: Yes

Reviewer #2: Yes

Reviewer #3: Yes

Reviewer #1: The authors need to carefully reorganize the manuscript, as there are numerous formatting mistakes throughout. However, the quality of the data presentation and the validity of the conclusions are sufficient for publication. Please revise the manuscript according to the comments provided in the attached file.

Reviewer #2: Overall Evaluation

This study utilized a cross-sectional design to compare the predictive efficacy of the TyG index and TyG-BMI index for non-alcoholic fatty liver disease (NAFLD) in elderly diabetic patients. With a relatively large sample size (n=6,882) and methodologically sound design, the findings suggest that the TyG-BMI index outperforms the TyG index in clinical utility. The research topic aligns with the current demand for non-invasive diagnostic tools. However, several methodological details require further clarification, and the discussion could benefit from deeper exploration.

Recommendations

1. Limitations of Ultrasound for NAFLD Diagnosis

The lower sensitivity of ultrasound may lead to potential underdiagnosis of NAFLD cases. It is recommended to supplement the Discussion section with an acknowledgment of this bias and its impact on the study’s conclusions.

2. Biological Interpretation of TyG-BMI

The cutoff values for TyG-BMI (212.886 and 251.741) should be contextualized within clinical frameworks. Comparisons with cutoffs from prior studies (e.g., Huet al.) are encouraged to elucidate discrepancies and explore underlying reasons.

3. Control of Confounding Factors

While Model 3 adjusted for liver enzymes (AST/ALT), the manuscript omits mention of other metabolic markers (e.g., HbA1c, insulin levels). Please clarify whether these variables were assessed for potential confounding effects or discuss limitations accordingly.

4. Depth of Discussion

Strengthen the Discussion by addressing the clinical implications and translational significance of the findings. Examples include:

- Is the TyG-BMI cutoff applicable to other ethnicities or regions?

- How can this tool be integrated into existing NAFLD screening workflows (e.g., as a replacement for or complement to ultrasound)?

5. Enrichment of Conclusions

Emphasize scenarios where the TyG-BMI index holds particular promise for implementation, such as primary care screening, to underscore its clinical relevance.

6. Ethics Statement Clarification

The ethics declaration cites "previous data and cross-sectional design" to justify exemption from informed consent. However, the degree of data anonymization and original consent scope for future research use remain unclear. It is recommended to supplement whether the data is completely anonymous (cannot be traced back to individuals), and to confirm whether the original data collection included consent clauses for future research purposes.

7.English Grammar

Enhance grammatical accuracy throughout the manuscript. For example, the lines 28-29 in the abstract (The AUCs of TyG and TyG-BMI indicated that both had predictive value for NAFLD, with TyG-BMI showing the highest predictive accuracy) contains a superlative error. As only two indices are compared, "higher" is more appropriate than "highest."

Reviewer #3: This study investigates the predictive value of two indices (TyG and TyG-BMI) for non-alcoholic fatty liver disease (NAFLD) in elderly diabetic patients. The topic is timely and relevant, especially given the increasing prevalence of NAFLD and metabolic syndrome in aging populations. However, the manuscript would benefit from improvements in clarity and scientific rigor in several sections. Below are my comments.

Figures and tables should be more clearly labeled. For instance, define all abbreviations in each figure/table legend. The logistic regression models need more detailed reporting—specifically about multicollinearity, confounding variables, and model fit. ROC curves and AUCs are presented but should be accompanied by confidence intervals and statistical comparisons (e.g., DeLong test).

Ultrasound is commonly used to detect fatty liver but may not reliably identify mild steatosis. Is it accurate enough to diagnose NAFLD in this study population?

Several types of diabetes medications have shown promise in improving NAFLD, particularly for those with type 2 diabetes. Pioglitazone stands out as a well-studied option, with strong evidence of improvement in liver histology, especially in patients with biopsy-proven NASH. Additionally, GLP-1RAs and SGLT2is have also demonstrated beneficial effects on NAFLD, including potential reductions in liver fat and improved liver enzymes. Have the authors reported the medication histories of all participants?

As noted in the discussion, “The TyG-BMI index, developed by Wang et al. in a Japanese population, has been shown to predict NAFLD effectively.” However, this study focuses on elderly patients with diabetes. It is unclear what new insights this research contributes beyond existing literature on TyG indices and NAFLD.

The phrase “Error! Reference source not found” appears multiple times in the manuscript. Please review the document carefully and correct these errors.

**Do you want your identity to be public for this peer review?** For information about this choice, including consent withdrawal, please see our Privacy Policy

Reviewer #1: No

Reviewer #2: No

Reviewer #3: No

---

## [Author Response · Author response to Decision Letter 1]

22 Jun 2025

Reviewer #1: The authors need to carefully reorganize the manuscript, as there are numerous formatting mistakes throughout. However, the quality of the data presentation and the validity of the conclusions are sufficient for publication. Please revise the manuscript according to the comments provided in the attached file.

Summary:

In this cross-sectional study, the author evaluated the effectiveness of the triglyceride-glucose index (TyG) and triglyceride-glucose-body mass index (TyG-BMI) in identifying non-alcoholic fatty liver disease (NAFLD) through a clinical study containing 6,882 elderly patients with diabetes mellitus (DM). Participants were divided into training and validation groups to assess diagnostic performance. Both TyG and TyG-BMI were significantly associated with NAFLD risk, with TyG-BMI showing believable predictive accuracy. In addition, Receiver-operating characteristic curve (AUROC) and Decision Curve Analysis (DCA) confirmed TyG-BMI as a more reliable clinical marker. Two TyG-BMI cutoffs were proposed: <212.886 to rule out NAFLD (sensitivity ~80%, NPV ~77%) and ≥251.741 to rule in NAFLD (specificity ~91%, PPV ~76%), consistent across both groups. Thus, these findings suggest TyG-BMI is a simple, non-invasive, and cost-effective method for NAFLD detection and classification in elderly diabetic populations. However, several issues still need to be addressed.

Comments:

1. In line 47-28, "when steatosis is less than 20% or in morbidly obese individuals", I think "especially when hepatic fat infiltration is below 20% or in cases of morbid obesity." is more suitable.

Answer: Thank you for your valuable suggestion. We have revised the sentence accordingly to improve clarity and readability. The updated sentence now reads:"Moreover, ultrasound may fail to detect hepatic steatosis, especially when hepatic fat infiltration is below 20% or in cases of morbid obesity." in Line 47-48.

2. In line 61-62, "this demographic" could be better to specify "elderly patients with diabetes" again for clarity.

Answer: Thank you for your insightful comment. To enhance clarity, we have revised the phrase “this demographic” to “elderly patients with diabetes” in Line 63-64 as suggested. We appreciate your careful review.

3. The author needs to carefully check their manuscript, for example, line 136, 213, 214, the Error! Reference source not found. And The figure legends should be organized well in a divided part.

Answer: Thank you for pointing this out. We have carefully reviewed the manuscript and removed all instances of “Error! Reference source not found.” including those in lines 136, 213, and 214. Additionally, the figure legends have been reorganized and clearly divided to improve clarity. We appreciate your helpful feedback.

4. The selected participants who had undergone liver ultrasound. This could introduce selection bias, because people undergoing liver ultrasound might already have suspected liver/metabolic issues.

Answer: Thank you for your insightful comment. We understand the concern regarding potential selection bias due to the inclusion of participants who underwent liver ultrasonography. However, we would like to clarify that the study population was derived from a large cohort of individuals undergoing routine physical examinations, not limited to those with known or suspected liver or metabolic diseases. Therefore, the included participants consisted of both individuals with potential liver abnormalities and those without any known issues. This approach helps reduce the likelihood of significant selection bias and enhances the generalizability of our findings.

5. The author defines DM as FBG >7.0 mmol/L or history of DM. However, this excludes patients diagnosed by HbA1c ≥6.5% or 2-hour OGTT ≥11.1 mmol/L according to ADA guidelines.

Answer: Thank you for your thoughtful comment. We acknowledge that the American Diabetes Association (ADA) guidelines include additional diagnostic criteria for diabetes mellitus (DM), such as HbA1c ≥6.5% or 2-hour plasma glucose ≥11.1 mmol/L following an oral glucose tolerance test (OGTT). However, in this retrospective study, we followed the approach used in several previous studies, where DM was defined based on fasting blood glucose (FBG) >7.0 mmol/L or a documented history of diabetes. For example, similar definitions were applied in the studies by Hu et al. (2022) [1] and Wang et al. (2021) [2], which focused on large-scale population-based screening for NAFLD and related metabolic disorders.

References:

1.Hu H, Han Y, Cao C, He Y. The triglyceride glucose-body mass index: a non-invasive index that identifies non-alcoholic fatty liver disease in the general Japanese population. J Transl Med. 2022;20(1):398.

2.Wang R, Dai L, Zhong Y, Xie G. Usefulness of the triglyceride glucose-body mass index in evaluating nonalcoholic fatty liver disease: insights from a general population. Lipids Health Dis. 2021;20(1):77.

6. In line 66, "cross-section study" should be "cross-sectional study"; In line 66 "waved" should be “waived”; In line 79, "flowchart of the selection of participants are shown" should be "is shown".

Answer: Thank you for carefully reviewing the manuscript and identifying these language errors. We have corrected “cross-section study” to “cross-sectional study”, and revised the sentence to “the flowchart of the selection of participants is shown” as suggested. “waved” was deleted and “the Ethics Committee granted a waiver of informed consent” was applied instead. We appreciate your attention to detail.

Reviewer #2: Overall Evaluation

This study utilized a cross-sectional design to compare the predictive efficacy of the TyG index and TyG-BMI index for non-alcoholic fatty liver disease (NAFLD) in elderly diabetic patients. With a relatively large sample size (n=6,882) and methodologically sound design, the findings suggest that the TyG-BMI index outperforms the TyG index in clinical utility. The research topic aligns with the current demand for non-invasive diagnostic tools. However, several methodological details require further clarification, and the discussion could benefit from deeper exploration.

Recommendations

1. Limitations of Ultrasound for NAFLD Diagnosis

The lower sensitivity of ultrasound may lead to potential underdiagnosis of NAFLD cases. It is recommended to supplement the Discussion section with an acknowledgment of this bias and its impact on the study’s conclusions.

Answer: We appreciate the reviewer’s insightful comment. In response, we have revised the Discussion section to acknowledge the potential limitations of using ultrasound for NAFLD diagnosis. Specifically, we added the following statement:

"Although ultrasound examination has been widely used for diagnosing NAFLD, its relatively low sensitivity may lead to underdiagnosis of NAFLD cases. This limitation could introduce selection bias or information bias into the study, thereby affecting the accuracy and reliability of the results." highlighted in Line 293-294

This addition aims to clarify the potential impact of diagnostic limitations on our findings and to enhance the transparency of our study.

2. Biological Interpretation of TyG-BMI

The cutoff values for TyG-BMI (212.886 and 251.741) should be contextualized within clinical frameworks. Comparisons with cutoffs from prior studies (e.g., Huet al.) are encouraged to elucidate discrepancies and explore underlying reasons.

Answer: Thank you for your valuable suggestion. We have compared our TyG-BMI cutoffs (212.886 and 251.741) with those from prior studies (e.g., Hu et al and Wang el al.) in the revised manuscript. We have analyzed potential discrepancies and explore underlying reasons in the Discussion section.

Hu et al. previously demonstrated in a Japanese cohort that two TyG-BMI cutoff values (182.2 and 224.0) could accurately identify NAFLD and reduce unnecessary ultrasonography in the population [(4)]. Xuejie Wang et al. further determined the optimal TyG-BMI cutoff value of 218.08 in a Chinese general population. Notably, while the cutoff values in our study were higher than those reported in the Japanese population, they showed similarity to the thresholds established in Chinese populations. (View in Discussion section in Iine 268-273 )

3. Control of Confounding Factors

While Model 3 adjusted for liver enzymes (AST/ALT), the manuscript omits mention of other metabolic markers (e.g., HbA1c, insulin levels). Please clarify whether these variables were assessed for potential confounding effects or discuss limitations accordingly.

Answer: Thank you for your critical insight regarding the inclusion of metabolic markers in our analysis. In the revised manuscript, we have acknowledged the omission of key metabolic markers such as HbA1c and insulin levels due to data limitations. We have added a statement in the Discussion section to clarify that these variables were not included in the analysis, which may limit the comprehensive adjustment of confounders. Additionally, we have emphasized our plan to collect these important metabolic parameters in future studies to further improve the predictive model.

Finally, some relevant metabolic indicators, such as glycated hemoglobin (HbA1c) and direct measures of insulin resistance, were not included due to data limitations, which may affect the comprehensive adjustment of confounders and the model’s predictive accuracy, and we plan to collect these important metabolic parameters to further improve the predictive model in the future.(View in Limitation of Discussion Section )

4. Depth of Discussion

Strengthen the Discussion by addressing the clinical implications and translational significance of the findings. Examples include:

- Is the TyG-BMI cutoff applicable to other ethnicities or regions?

- How can this tool be integrated into existing NAFLD screening workflows (e.g., as a replacement for or complement to ultrasound)?

Answer: Thank you for the insightful suggestion. We have revised the Discussion to highlight the clinical relevance of TyG-BMI. As a simple, non-invasive, and cost-effective tool based on routine clinical data, TyG-BMI may complement or triage NAFLD screening in primary care, especially where imaging resources are limited. In our cohort, a TyG-BMI ≥251.741 showed high specificity (91.5%) for identifying high-risk individuals, while <212.886 showed high sensitivity (80.0%) for ruling out NAFLD. Although derived from a Chinese elderly diabetic population, similar values reported in other studies suggest broader applicability, though further validation across ethnicities is needed. Integration into EHR systems may support early detection and monitoring in clinical workflows.(View in Discussion Section in Iine 291-295)

5. Enrichment of Conclusions

Emphasize scenarios where the TyG-BMI index holds particular promise for implementation, such as primary care screening, to underscore its clinical relevance.

Answer: In response, we have added a clarification highlighted in Line 304-308:”In conclusion, although the prevalence of NAFLD remained high among elderly individuals with diabetes, a stepwise decline was observed with increasing age in both elderly males and females. In this population, the TyG-BMI index demonstrated superior predictive accuracy for NAFLD compared to the TyG index alone. Given its simplicity, non-invasiveness, and low cost, the TyG-BMI index holds particular promise for implementation in primary care settings as a practical screening tool to identify elderly diabetic patients at high risk for NAFLD, thereby underscoring its clinical utility in early detection and prevention strategies.”

6. Ethics Statement Clarification

The ethics declaration cites "previous data and cross-sectional design" to justify exemption from informed consent. However, the degree of data anonymization and original consent scope for future research use remain unclear. It is recommended to supplement whether the data is completely anonymous (cannot be traced back to individuals), and to confirm whether the original data collection included consent clauses for future research purposes.

Answer: In the revised manuscript , we have already made modifications in Line 67-73 to address the Ethics Statement clarification.

7.English Grammar

Enhance grammatical accuracy throughout the manuscript. For example, the lines 28-29 in the abstract (The AUCs of TyG and TyG-BMI indicated that both had predictive value for NAFLD, with TyG-BMI showing the highest predictive accuracy) contains a superlative error. As only two indices are compared, "higher" is more appropriate than "highest."

Answer: Thank you for the reviewer’s suggestion. The use of "highest" in the original manuscript was inappropriate due to our oversight, and it has been corrected to "higher" in Line 28.

Reviewer #3: This study investigates the predictive value of two indices (TyG and TyG-BMI) for non-alcoholic fatty liver disease (NAFLD) in elderly diabetic patients. The topic is timely and relevant, especially given the increasing prevalence of NAFLD and metabolic syndrome in aging populations. However, the manuscript would benefit from improvements in clarity and scientific rigor in several sections. Below are my comments.

1. Figures and tables should be more clearly labeled. For instance, define all abbreviations in each figure/table legend. The logistic regression models need more detailed reporting—specifically about multicollinearity, confounding variables, and model fit. ROC curves and AUCs are presented but should be accompanied by confidence intervals and statistical comparisons (e.g., DeLong test).

Answer: Thank you for this helpful suggestion. In accordance with your suggestions, we have implemented the following revisions:

(1) We have revised all figure and table legends to include definitions of all abbreviations (View in Tables and figures)

(2) We have added a section to the Results (or Methods) section detailing the assessment of multicollinearity, including Variance Inflation Factors (VIFs) for all predictor variables. In addition, we have included additional measures of model fit (Akaike Information Criterion) to better demonstrate the adequacy of the model. Specifically as follows:

Utilizing the Akaike Information Criterion (AIC) to assess the relative fit of competing models, a lower AIC value indicates a superior model in terms of parsimony and explanatory power. Variance Inflation Factors (VIFs) were calculated to assess multicollinearity among the independent variables in the model, and a VIF greater than 5 is commonly considered indicative of high multicollinearity.(View in Statistical analysis of Methods section)

Otherwise, the decreasing AIC values across the models suggest a progressively improved model fit, indicating a better balance between model complexity and explanatory power for the observed data (Table 2). A VIF value of less than 5 for each variable in the model suggests that multicollinearity is not a concern(Supplemental Table 1) (View in ‘Association of TyG and TyG-BMI with NAFLD in elderly participants with diabetes’ of Results section)

(3) The previous version already included confidence intervals for the AUCs and statistical comparisons. To enhance clarity, we have added corresponding annotations to the table 4. The specific revisions are as follows:

The predictive abilities of TyG and TyG-BMI for NAFLD were evaluated using the area under the receiver operating characteristic (AUROC) curves. The AUROC values were compared using the Delong test.(View in Statistical analysis of Methods section)

Table 4 The optimal cutoff point for the triglyceride-glucose-body mass index and triglyceride-glucose in diagnosing non-alcoholic fatty liver disease in training and validation group.

Dataset Marker CutOff AUC (95%CI) P value* Sensitivity Specificity PPV NPV

Training TyG-BMI 219.028 0.771(0.758-0.784) Reference 0.748 0.657 0.649 0.754

TyG 8.929 0.678(0.663-0.692) <0.001 0.690 0.585 0.585 0.690

Validation TyG-BMI 215.712 0.756(0.738-0.779) Reference 0.770 0.626 0.641 0.759

TyG 9.109 0.703(0.680-0.725) <0.001 0.586 0.723 0.646 0.668

AUC, area under curve; PPV, positive predictive value; NPV, negative predictive value; TyG-BMI, triglyceride-glucose-body m

---

## [Decision Letter · Decision Letter 1]

22 Jul 2025

Dear Dr. Cai,

Thank you for submitting your manuscript to PLOS ONE. After careful consideration, we feel that it has merit but does not fully meet PLOS ONE’s publication criteria as it currently stands. Therefore, we invite you to submit a revised version of the manuscript that addresses the points raised during the review process.

Please make peer-to-peer modifications to the reviewer's comments.

We look forward to receiving your revised manuscript.

Kind regards,

Qian Wu

Academic Editor

PLOS ONE

Journal Requirements:

Reviewers' comments:

Reviewer's Responses to Questions

**Comments to the Author**

Reviewer #4: All comments have been addressed

Reviewer #5: (No Response)

Reviewer #6: All comments have been addressed

2. Is the manuscript technically sound, and do the data support the conclusions?

Reviewer #4: Yes

Reviewer #5: Partly

Reviewer #6: Yes

3. Has the statistical analysis been performed appropriately and rigorously?

Reviewer #4: Yes

Reviewer #5: Yes

Reviewer #6: Yes

4. Have the authors made all data underlying the findings in their manuscript fully available?

Reviewer #4: Yes

Reviewer #5: Yes

Reviewer #6: Yes

5. Is the manuscript presented in an intelligible fashion and written in standard English?

Reviewer #4: Yes

Reviewer #5: No

Reviewer #6: Yes

Reviewer #4: (No Response)

Reviewer #5: This manuscript entitled « Comparison of the triglyceride-glucose index and triglyceride-glucose-body mass index for predicting non-alcoholic fatty liver disease in elderly diabetic patients » was sent by Zhu et al. for a reviewing process to Plos One journal.

The aim of this study was to highlight a prospective and non invasive tool, the triglyceride-glucose-body mass index (TyG-BMI), to predict non-alcoholic fatty liver disease with the help of a large cohort of 6,882 individuals aged at least 60 years old and with diabetes mellitus.

The introduction and the methodology part are good and clearly present the subject with the dedicated references.

The results part requires and in depth work to constrictively present the results and the dedicated references, and to allow targeted comments for reviewing such as a comment on the fact that NAFLD decreases gradually as subjects aged, or why the three different models authors have used are not subsequently studied. The layout should be also redone. The quality of the figures should be improved.

The discussion is interesting.

Overall, the topic of this study is interesting and clearly addressed at the start of the manuscript. But, the linearity of the results is interrupted by layout problems and clarity.

Reviewer #6: please address the issues mentioned in the attachment. Especially some of the nomenclature used need to be changed. NFALD present vs absent. Non- NFALD does not seem appropriate

**Do you want your identity to be public for this peer review?** For information about this choice, including consent withdrawal, please see our Privacy Policy

Reviewer #4: No

Reviewer #5: No

Reviewer #6: **Yes:** Hari Naga Garapati

---

## [Author Response · Author response to Decision Letter 2]

1 Sep 2025

Reviewer 4:

Reviewer 5:

The results part requires and in depth work to constrictively present the results and the dedicated references, and to allow targeted comments for reviewing such as a comment on the fact that NAFLD decreases gradually as subjects aged, or why the three different models authors have used are not subsequently studied. The layout should be also redone. The quality of the figures should be improved.

Answer: Thanks for your careful advice. We submitted the Supplemental Table S1 to show the relationship between the prevalence of NAFLD and age. And due to the length limitation and research objective of the paper, there was no further analysis on the low incidence rate of NAFLD among the elderly. In future, we will conduct the epidemiologic study of NAFLD in the eldly. And we have added the relevant content at the end of the Discussion Part in Line 302-303. Furthermore, the three models just proved that adjusting different types of variables could make the model stable, so we did not conduct further analysis and research. We have adjusted the position of Table 2 in Line 156. And We have uploaded the clear figures again.

Reviewer 6:

1) Abstract: Line 17:

Developing an effective, simple, noninvasive method to identify NAFLD, track disease progression, and monitor treatment effects is importance- correction – important.

Line 17 and 18: This study aims to assess the value of triglyceride-glucose index (TyG) and triglyceride-glucose-body mass index (TyG-BMI) in detecting NAFLD elderly patients with diabetes mellitus (DM). correction to - NAFLD in elderly patients.

Answer: Thanks for your kind reminder. “NAFLD in elderly diabetic patients” is more suitable in this part. We have changed this content in Line 17.

2) Table 1- suggest change of Nomenclature- use presence and absence of NAFLD- use of non-NAFLD is confusing and almost represents as Alcoholic fatty liver disease.

Answer: Thank you for your careful reminder. We have changed the Table title to “Baseline characteristics for the training and validation groups by NAFLD or non-NAFLD status in diabetes population.”

3) Reference sources needed- showing as errors?- need explanation as to why they are still being used that way.

Answer: Thank you for pointing out our mistakes. Due to unfamiliarity with the literature citation software, there were some errors in the manuscript. We have double checked and resolved them.

---

## [Decision Letter · Decision Letter 2]

22 Sep 2025

Dear Dr. Cai,

Thank you for submitting your manuscript to PLOS ONE. After careful consideration, we feel that it has merit but does not fully meet PLOS ONE’s publication criteria as it currently stands. Therefore, we invite you to submit a revised version of the manuscript that addresses the points raised during the review process.

We look forward to receiving your revised manuscript.

Kind regards,

Qian Wu

Academic Editor

PLOS ONE

Journal Requirements:

Additional Editor Comments:

Reviewer #5:

Reviewer #6:

Reviewers' comments:

Reviewer's Responses to Questions

**Comments to the Author**

Reviewer #5: (No Response)

Reviewer #6: All comments have been addressed

2. Is the manuscript technically sound, and do the data support the conclusions?

Reviewer #5: Yes

Reviewer #6: Yes

3. Has the statistical analysis been performed appropriately and rigorously?

Reviewer #5: I Don't Know

Reviewer #6: Yes

4. Have the authors made all data underlying the findings in their manuscript fully available?

Reviewer #5: Yes

Reviewer #6: Yes

5. Is the manuscript presented in an intelligible fashion and written in standard English?

Reviewer #5: Yes

Reviewer #6: Yes

Reviewer #5: Zhu et al. sent a manuscript entitled « Comparison of the triglyceride-glucose index and triglyceride-glucose-body mass index for predicting non-alcoholic fatty liver disease in elderly diabetic patients», for a second round of a reviewing process to Pone.

The aim of this study was to highlight a prospective and non invasive tool, the triglyceride-glucose-body mass index (TyG-BMI), to predict non-alcoholic fatty liver disease with the help of a large cohort of 6,882 individuals aged at least 60 years old and with diabetes mellitus.

The introduction and the methodology part are still good and clearly present the subject with the dedicated references.

But, the results part remains insufficient and requires and in depth work to constructively present the results and the dedicated references, and to allow targeted comments for reviewing such as a comments on the fact that NAFLD seems to decrease gradually as subjects aged, or why the three different models authors have used are not subsequently studied. In addition for this last point, authors should justify the choice of the parameters in each model (with the dedicated references). This lack of organisation could be easily put right and might strengthen the impact of the results of the authors.

Line 85 Error reference

Order of the figures

Table 1, authors should emphasize the differences plotted between the two groups.

Line 152 and 254 layout

Line 183 layout Figure title

Line 216 and 268 « cutoff »

LIne 282 de novo « italic »

The quality of the figures should be improved.

Reviewer #6: (No Response)

**Do you want your identity to be public for this peer review?** For information about this choice, including consent withdrawal, please see our Privacy Policy

Reviewer #5: No

Reviewer #6: **Yes:** Hari Naga Garapati

---

## [Author Response · Author response to Decision Letter 3]

22 Oct 2025

Answer:

Thanks for your careful advice. We have a comment about the fact that NAFLD seems to decrease gradually as subjects aged in Line 240-246 and we will analyze the impact of age on NAFLD in the elderly in future research due to the length limitation and research direction of this paper. The three models with the decreasing AIC values and a VIF value of less than 5 were used to prove that adjusting different types of variables could make the model stable, so we did not conduct further studies. As to the choice of the parameters in each model, we selected only those parameters that demonstrated statistical significance in univariate analysis based on the literature (Wang R, Dai L, Zhong Y, et al. Usefulness of the triglyceride glucose-body mass index in evaluating nonalcoholic fatty liver disease: insights from a general population. Lipids Health Dis, 2021, 20(1): 77) and then divided them into general demographic data and the general examination items in our hospital, like complete blood count and liver-kidney glyco-lipid test. We also modify the annotation below Table 2 in Line 157-159 to more clearly indicate the adjustment of each model variable.

At last, we modify the “Error reference” in Line 85, change the order of the figures and figure title in Line 86, 176, 184 and 235. Apart from LDL-C, there are no significant differences between the training and validation groups, which indicates that the data has the same distribution between the training and validation groups to achieve more stable and accurate results later. We separately expressed the significant differences within the training and validation sets in Line 141-145. We also changed the “cutoff” and “de novo” with the correct format.

The quality of the figures has been reuploaded, and we found that the higher clarity of the figures need to be downloaded instead of in submission view. So we uploaded the PDF and TIFF format together.

---

## [Decision Letter · Decision Letter 3]

11 Nov 2025

Dear Dr. Cai,

Thank you for submitting your manuscript to PLOS ONE. After careful consideration, we feel that it has merit but does not fully meet PLOS ONE’s publication criteria as it currently stands. Therefore, we invite you to submit a revised version of the manuscript that addresses the points raised during the review process.

We look forward to receiving your revised manuscript.

Kind regards,

Qian Wu

Academic Editor

PLOS ONE

Journal Requirements:

Reviewers' comments:

Reviewer's Responses to Questions

**Comments to the Author**

Reviewer #5: (No Response)

2. Is the manuscript technically sound, and do the data support the conclusions?

Reviewer #5: Yes

3. Has the statistical analysis been performed appropriately and rigorously?

Reviewer #5: Yes

4. Have the authors made all data underlying the findings in their manuscript fully available?

Reviewer #5: Yes

5. Is the manuscript presented in an intelligible fashion and written in standard English?

Reviewer #5: Yes

Reviewer #5: Zhu et al. sent a manuscript entitled « Comparison of the triglyceride-glucose index and triglyceride-glucose-body mass index for predicting non-alcoholic fatty liver disease in elderly diabetic patients», for a third round of a reviewing process to Pone.

The aim of this study is to highlight a prospective and non invasive tool, the triglyceride-glucose-body mass index (TyG-BMI), to predict non-alcoholic fatty liver disease in a large cohort of 6,882 individuals aged at least 60 years old and with diabetes mellitus.

As mentioned previously, the description of the results present a lack a clarity, and still require an in depth re-organisation and presentation.

Some additional typo comments

Line 27/81 remove « space »

Line 86

Cut-off or cutoff

Line 230 Fig.4 —> Figure 4

Line 235 FLD —> NAFLD

LIne 261, 264, 265 et al. (Italic)

**Do you want your identity to be public for this peer review?** For information about this choice, including consent withdrawal, please see our Privacy Policy

Reviewer #5: No

---

## [Author Response · Author response to Decision Letter 4]

14 Dec 2025

Answer:

We sincerely thank the reviewer for these insightful and constructive comments, which have substantially improved the scientific rigor and clarity of our manuscript. We have carefully addressed each concern raised, and the specific revisions are detailed below:

(1) Explanation for age-related decline in NAFLD prevalence

We thank the reviewer for highlighting this important finding. This finding can be attributed to the age-related physiological changes or the influence of survivor bias. We have added a dedicated discussion of this phenomenon in the Discussion section, Lines 267-270.

(2) Justification for the three hierarchical models with dedicated references

We fully agree with the reviewer's concern. The covariates included in our three hierarchical models were selected systematically based on established literature and prior evidence of their associations with NAFLD. The AIC values were used to compare the relative fit of the three models and to determine which set of covariates provided the optimal balance between explanatory power. Additionally, it should be noted that the predictive ability and reclassification statistics, as well as the clinical utility analysis, were all performed specifically for the optimal predictive models (Model 3) with TyG/TyG-BMI according to AIC values. We have substantially revised the Methods section to provide a comprehensive justification for our hierarchical modeling strategy, supported by dedicated references. We made the following revisions in the Methods, Line 109-122 and Results sections, Line 161-162.

This stepwise approach allows readers to assess the independence and robustness of the associations across different levels of covariate adjustment, which is a recommended practice in epidemiological research.

(3) Enhanced presentation and justification of subgroup analyses

We appreciate this comment and have substantially enhanced the Results section to provide more detailed justification and clearer presentation of our stratified analyses. We have implemented corresponding revisions in the Methods, Line 125-127, Results, Line 174-178 and Table 3, Line 182-185.

(4) Modification of some additional errors

We are grateful for the careful advice. We remove the « space » in Line 27 and 81. The “Fig. 4” has been revised as “Figure 4” in Line 244. We modify the caption and legend of Fig. 4 in Line 250-252 and change the corresponding content mentioned above. At last, we change all instances of “et al.” to italics.

---

## [Decision Letter · Decision Letter 4]

4 Jan 2026

Comparison of the triglyceride-glucose index and triglyceride-glucose-body mass index for predicting non-alcoholic fatty liver disease in elderly diabetic patients

PONE-D-25-17049R4

Dear Dr. Cai,

We’re pleased to inform you that your manuscript has been judged scientifically suitable for publication and will be formally accepted for publication once it meets all outstanding technical requirements.

Kind regards,

Qian Wu

Academic Editor

PLOS One

Additional Editor Comments (optional):

Reviewers' comments:

Reviewer's Responses to Questions

**Comments to the Author**

Reviewer #5: All comments have been addressed

2. Is the manuscript technically sound, and do the data support the conclusions?

Reviewer #5: Yes

3. Has the statistical analysis been performed appropriately and rigorously?

Reviewer #5: Yes

4. Have the authors made all data underlying the findings in their manuscript fully available?

Reviewer #5: Yes

5. Is the manuscript presented in an intelligible fashion and written in standard English?

Reviewer #5: Yes

Reviewer #5: Zhu et al. sent a manuscript entitled « Comparison of the triglyceride-glucose index and triglyceride-glucose-body mass index for predicting non-alcoholic fatty liver disease in elderly diabetic patients», for a final round of a reviewing process to Pone.

Authors properly answered and completed the manuscript following reviewers’ instructions. There is no additional comments regarding the publication of this manuscript.

**Do you want your identity to be public for this peer review?** For information about this choice, including consent withdrawal, please see our Privacy Policy

Reviewer #5: No

---

## [Editor Report · Acceptance letter]

PONE-D-25-17049R4

PLOS One

Dear Dr. Cai,

I'm pleased to inform you that your manuscript has been deemed suitable for publication in PLOS One. Congratulations! Your manuscript is now being handed over to our production team.

Kind regards,

on behalf of

Dr. Qian Wu

Academic Editor

PLOS One